# ONC201 Suppresses Neuroblastoma Growth by Interrupting Mitochondrial Function and Reactivating Nuclear ATRX Expression While Decreasing MYCN

**DOI:** 10.3390/ijms24021649

**Published:** 2023-01-13

**Authors:** Jian-Ching Wu, Chao-Cheng Huang, Pei-Wen Wang, Ting-Ya Chen, Wen-Ming Hsu, Jiin-Haur Chuang, Hui-Ching Chuang

**Affiliations:** 1Center for Mitochondrial Research and Medicine, Kaohsiung Chang Gung Memorial Hospital, Kaohsiung 83301, Taiwan; 2Department of Pathology, Kaohsiung Chang Gung Memorial Hospital and Chang Gung University College of Medicine, Kaohsiung 83301, Taiwan; 3Biobank and Tissue Bank, Kaohsiung Chang Gung Memorial Hospital, Kaohsiung 83301, Taiwan; 4Department of Internal Medicine, Kaohsiung Chang Gung Memorial Hospital and Chang Gung University College of Medicine, Kaohsiung 83301, Taiwan; 5Department of Surgery, National Taiwan University Hospital, National Taiwan University College of Medicine, Taipei 10617, Taiwan; 6Department of Pediatric Surgery, Kaohsiung Chang Gung Memorial Hospital and Chang Gung University College of Medicine, Kaohsiung 83301, Taiwan; 7Department of Otolaryngology, Kaohsiung Chang Gung Memorial Hospital and Chang Gung University College of Medicine, Kaohsiung 83301, Taiwan

**Keywords:** neuroblastoma, ATRX, MYCN, ClpPX protease, ONC201, apoptosis, mitochondria

## Abstract

Neuroblastoma (NB) is characterized by several malignant phenotypes that are difficult to treat effectively without combination therapy. The therapeutic implication of mitochondrial ClpXP protease ClpP and ClpX has been verified in several malignancies, but is unknown in NB. Firstly, we observed a significant increase in ClpP and ClpX expression in immature and mature ganglion cells as compared to more malignant neuroblasts and less malignant Schwannian-stroma-dominant cell types in human neuroblastoma tissues. We used ONC201 targeting ClpXP to treat NB cells, and found a significant suppression of mitochondrial protease, i.e., ClpP and ClpX, expression and downregulation of mitochondrial respiratory chain subunits SDHB and NDUFS1. The latter was associated with a state of energy depletion, increased reactive oxygen species, and decreased mitochondrial membrane potential, consequently promoting apoptosis and suppressing cell growth of NB. Treatment of NB cells with ONC201 as well as the genetic attenuation of ClpP and ClpX through specific short interfering RNA (siRNA) resulted in the significant upregulation of the tumor suppressor alpha thalassemia/mental retardation X-linked (ATRX) and promotion of neurite outgrowth, implicating mitochondrial ClpXP proteases in *MYCN*-amplified NB cell differentiation. Furthermore, ONC201 treatment significantly decreased MYCN protein expression and suppressed tumor formation with the reactivation of ATRX expression in *MYCN*-amplified NB-cell-derived xenograft tumors. Taken together, ONC201 could be the potential agent to provide diversified therapeutic application in NB, particularly in NB with *MYCN* amplification.

## 1. Introduction

Neuroblastoma (NB) is a genetically heterogeneous tumor characterized by pleomorphic cells, including those with *MYCN* amplification that are highly glycolytic, presenting the opportunity to use glycolytic inhibitor 2-deoxyglucose (2DG) to suppress NB cell growth [1,2]. Significant changes in mitochondrial membrane potential, impaired mitochondrial function, increased oxidative stress, and subsequent cell death are involved in the process [3,4], indicating the importance of studying mitochondrial function.

The proper execution of mitochondrial functions depends on the maintenance of the integrated mitochondrial proteome. Mitochondrial unfolded protein response (UPR^mt^) is a feedback loop, wherein the dysfunction of the mitochondrial proteome is sensed and communicated to the nucleus, which subsequently launches an extensive transcriptional program to repair the damage and to rescue mitochondrial function [5]. UPR^mt^ is maintained primarily by the chaperones HSP10, HSP60, HSP70, and HSP90 and by proteases of the AAA+ superfamily, specifically the Lon, ClpXP, and m-AAA proteases. Dysregulation of the proteases in cancer may serve as a novel therapeutic target in malignancies which are highly dependent on mitochondrial functions for survival, including acute myeloid leukemias and a subset of melanomas [6]. ClpXP protease is composed of ClpP, which forms a multimeric complex with ClpX, leading to proteolytic activity. ClpXP is overexpressed in hematologic malignancies and solid tumors. Notably, both the inhibition and hyperactivation of ClpXP was reported to lead to impaired respiratory chain activity and cause cancer cell death [7]. 

ONC201 was discovered initially as a first-in-class TNF-related apoptosis-inducing ligand (TRAIL) compound (known as TIC10). Based on the central core structure of ONC201, several compounds of the imipridone family have been developed, including ONC201, ONC206, and ONC212. There are more than 15 ongoing Phase I and Phase II clinical trials of ONC201, either as a single agent or in combination, to treat cancers including glioblastoma, acute leukemia, breast, colorectal, endometrial high-grade gliomas, multiple myeloma, neuroendocrine, and other solid cancers [8,9]. 

Many studies emphasize the importance of ClpP in mediating cancer cell death, including in breast cancer tissue, SK-N-SH neuroblastoma cells, and acute myeloid leukemia [10,11,12,13]. Moreover, the coordination of ClpP and ClpX was important in a study that showed boron-based peptidomimetics as potent inhibitors of human ClpP in the presence of human ClpX [10]. ClpX is suppressed upon ONC212 treatment in pancreatic cancer cells and in diffuse intrinsic pontine glioma (DIPG) cells treated with ONC206 [11,12]. However, both ClpP and ClpX are suppressed when DIPG cells are treated with ONC212 [12]. Results of these studies clearly indicate the importance of studying both ClpP and ClpX when treating cancer cells with imipridones because their diversified response is obviously dependent on the cell types as well as on different members of the imipridone family. 

In addition to cell proliferation and death, ClpXP protease is also involved in cell differentiation, including the morphological differentiation of Streptomyces lividans [13,14]. Preclinical testing of a combined treatment with the ClpP activator ONC201/TIC10 and 2DG resulted in a dual metabolic reprogramming and synergistic anti-proliferative and anti-migratory effects on glioblastoma cells [15]. In that respect, ClpP-mediated degradation of mitochondrial energy-related components is additive to the effects of 2DG as a glycolytic inhibitor [16]. However, our previous study has revealed that 2DG alone can simultaneously target cancer and endothelial cells to significantly suppress NB growth in mice regardless of the status of *MYCN* amplification [2]. The latter means that if we use 2DG in combination with ONC201 to treat NB cells, it might mask the efficacy of ONC201 for the treatment of NB. In this study, we decided to test the effect of ONC201 alone on NB cell differentiation and tumor growth. The results offer promise for the development of future therapy for NB.

## 2. Results

### 2.1. The Expressions of ClpP and ClpX Are Associated with NB Differentiation

Elevated ClpP expression is associated with poor survival in patients with other malignancies, but remains unclear with regard to NB tumors. Firstly, we evaluated the expression of ClpP and ClpX from the resected tumors of patients with NB. Immunostaining of ClpP and its multimeric partner ClpX was herein performed in NB tumors. A total of 23 patients with 5 different International Neuroblastoma Pathology Classification (INPC) statuses were included. The staining intensity of ClpP and ClpX varied significantly according to the cell type, even in different tissue blocks of the same patient. Accordingly, the expressions of ClpP and ClpX in the NB tumors were classified according to the predominant cell type, categorized as primitive neuroblast, differentiated neuroblast, immature ganglion cell, mature ganglion cell, and Schwannian stroma. The expressions of ClpP and ClpX were clearly different among the various cell types (Figure 1A). The primitive neuroblast and Schwannian stroma cell types were either devoid of ClpP expression or were very weak; on the contrary, the expression of ClpP in the immature ganglion cells was significantly higher than in the former two cell types, followed by that of mature ganglion cells and differentiated neuroblasts (Figure 1B). The expression of ClpX in the NB cells essentially followed the same trend as that of ClpP, except that the highest expression was identified in mature ganglion cells, followed by immature ganglion cells, and differentiated neuroblasts. The primitive neuroblast and Schwannian stroma were essentially devoid of ClpX expression (Figure 1C). These results suggest that both ClpP and ClpX may involve the differentiation process in NB.

### 2.2. *Administration of ONC201 Inhibits NB Cell Growth* and Promotes NB Cell Death

To investigate the efficacy of ONC201 alone in the treatment of NB cells, *MYCN*-nonamplified SK-N-AS and *MYCN*-amplified BE(2)M17 cell lines were treated with various concentrations of ONC201 (from 0 to 40 μM) for 48 h and 96 h. As shown in Figure 2A, treatment of ONC201 alone displayed about 20% of growth suppression in two NB cell lines at 48 h; however, more than 60% was suppressed in both cell lines at 96 h, suggesting that the low concentration of ONC201 is sustainable for the treatment of NB. The effect of ONC201 on the ClpXP complex in the four NB cell lines was measured by Western blot. Interestingly, the results of the short treatment period demonstrate that ONC201 (5 μM) did not affect the protein levels of ClpP, but was associated with the significant downregulation of ClpX expression in the four NB cell lines (Appendix A). For the longer treatment period, ONC201 almost eliminated the protein expression of ClpX at both 24 h and 48 h, whereas ClpP was notably decreased by ONC201 at 48 h in the four NB cell lines (Appendix A). The phase contrast images show that the cell density was markedly decreased with ONC201 (5 μM) treatment of SK-N-AS cells and BE(2)M17 cells (Figure 2B).

We further evaluated cell viability by trypan blue staining using flow cytometry analysis as well as TUNEL assay. Consistent with the previous results of cell growth inhibition, the percentage of trypan-blue-positive cells was less significant at 48 h of treatment with ONC201, but more significant at 96 h in both SK-N-AS and BE(2)M17 NB cells (Figure 2C).

An additional assay using TUNEL staining to detect DNA breaks also revealed significantly increased cell death in both NB cell lines treated with ONC201 for 48 h (Figure 2D,E). These results suggest that monotherapy with ONC201 alone is effective to suppress NB cell growth and promote cell death within the time frame of the study, regardless the status of *MYCN* amplification.

We also studied the effect of siRNA targeting ClpP and ClpX on the proliferation and death of SK-N-AS and BE(2)M17 cells. The data show that siClpP and sClpX could significantly decrease cell proliferation while increase cell death in both NB cells (Appendix A).

### 2.3. ONC201 Alone Is Capable of Disturbing Respiratory Chain Proteins and Mitochondrial Functions in NB Cells

Respiratory chain proteins have previously been identified as putative ClpP substrates, and the inhibition of ClpP is known to accumulate degraded or misfolded subunits. Thus, we herein investigated two representative respiratory chain subunits, SDHB and NDUFS1. The results reveal that treatment with ONC201 alone remarkably reduced ClpP and ClpX expressions in representative SK-N-AS cells (Figure 3A). Moreover, the protein levels of SDHB and NDUFS1 were significantly decreased by ONC201 in SK-N-AS cells (Figure 3A). We next investigated whether mitochondrial functions were changed in NB cells. To achieve this, the Seahorse XF analyzer was applied to evaluate OCR (quantity of mitochondrial respiration) and ECAR (indicator of glycolysis) in ONC201-treated NB cells. The OCR of SK-N-AS cells treated with ONC201 (5 μM) significantly diminished to less than 30 pMoles/min/2 × 10^4^ cells of basal respiration (Figure 3B,C). However, ONC201 treatment did not boost the inhibitory effect on ECAR (Figure 3D,E). These results reveal that ONC201 inhibits the oxidative phosphorylation chain but not glycolysis in the NB cells.

We next examined other mitochondrial functions, including MMP and ATP production, in NB cells treated with ONC201. The MMP detected by TMRM revealed that ONC201 treatment significantly decreased the MMP in both SK-N-AS and BE(2)17 cells (Figure 3F). The baseline ATP contents were higher in the *MYCN*-amplified BE(2)17 cells than in the *MYCN*-nonamplified SK-N-AS cells. However, single treatment with ONC201 significantly decreased of the ATP production in both NB cells (Figure 3G). Collectively, these results indicate that ONC201 treatment alone could disturb mitochondrial functions regardless of the status of *MYCN* amplification.

### 2.4. ONC201 Treatment Induces ROS Generation and Apoptosis in NB Cells

Excess ROS has been implicated in the regulation of apoptosis in several tumor types. We thus measured the ROS content by DHE and MitoSox Red assays in ONC201-treated NB cells. A significant increase in DHE fluorescence was detected in the ONC201-treated SK-N-AS and BE(2)17 cells (Figure 4A). Subsequently, MitoSox Red staining revealed an elevated mitochondrial ROS accumulation in ONC201-treated SK-N-AS and BE(2)17 cells (Figure 4B).

To investigate whether the treatment triggered mitochondria-mediated intrinsic apoptosis in NB cells, NB cells were treated with ONC201 for 48 h before being harvested for cytosol and mitochondria isolation. Western blot analysis indicated that ONC201 significantly increased the expression of cytochrome C in the cytosol and mitochondria fraction in both SK-N-AS and BE(2)17 cells (Figure 4C,D).

To further verify the apoptotic signaling activation, we evaluated the expression of cleaved caspase 3 by Western blot. The results reveal that ONC201 alone significantly enhanced the protein levels of cleaved caspase 3 in both SK-N-AS and BE(2)17 cells (Figure 4E). These results suggest that single treatment with ONC201 could provoke ROS generation and mitochondria-mediated apoptosis in NB cells, regardless of the status of *MYCN* amplification.

### 2.5. Administration of ONC201 Induces Neurite Outgrowth and Promotes ATRX Expression and Nuclear Translocation in MYCN-Amplified NB Cells

Via microscopy observations, we found that neurite outgrowth was observed when NB cells were treated with ONC201, which was obvious in BE(2)M17 cells, but not in SK-N-AS cells (Figure 5A). Given that ATRX has been reported to exhibit involvement in chromatin remodeling and neuronal differentiation [17], we assessed the levels of ATRX in NB cells when exposed to ONC201. Western blot analysis showed that ONC201 significantly increased the protein levels of ATRX in BE(2)M17 cells, which was associated with a significant decrease in MYCN (Figure 5B). To further investigate the distribution of increased ATRX in NB cells, immunofluorescence staining was performed to label ATRX (red) and mitochondria using an anti-TOM20 antibody (green) under confocal microscopy observation. Of note, stronger intensity of red fluorescence was found in the nuclei of ONC201-treated SK-N-AS and BE(2)M17 cells compared with the control group (Figure 5C). Consistent with the above findings, there was an increase in ATRX in the nucleus, but a decrease in ATRX in the cytosol (Figure 5D).

To investigate whether ClpXP downregulation can modulate ATRX expression, BE(2)M17 cells were transfected with siClpP or siClpX for 48 h and then harvested for Western blot analysis. The results indicate that the protein levels of ClpP and ClpX were reduced up to 50% in both siClpP- and siClpX-treated cells compared with the control group. As expected, the ATRX expression was significantly increased in BE(2)M17 cells receiving siClpP or siClpX, implying that ATRX upregulation might be dependent on the suppression of ClpXP protease (Figure 5E).

### 2.6. ONC201 Significantly Suppresses the Development of MYCN-Amplified NB Xenograft Tumors

Next, we evaluated the therapeutic efficacy of ONC201 in a xenograft NB animal model. Administration of ONC201 (50 μg/g) significantly inhibited the tumor growth. Compared to the control group, the tumor weight of ONC201-treated group (0.28 ± 0.27 g) decreased down to only around 20% of the control group (1.44 ± 0.45 g) (Figure 6A). Further immunohistochemistry analysis demonstrated a significantly higher percentage of nuclear ATRX-positive cells in ONC201-treated NB tissues than the control group (Figure 6B). Interestingly, the immunoreactive ClpP and ClpX protein staining was also significantly increased in the high-dose group, indicating that the administration of ONC201 (50 μg/g) for a longer period may reverse the decreased protein expression of both ClpP and ClpX in the in vitro studies (Figure 6C,D). We also verified the effect of ONC201 on MYCN expression in *MYCN*-amplified NB cells. Surprisingly, the ONC201 treatment suppressed the expression of MYCN in BE(2)M17 cells in a dose-dependent manner (Figure 6E).

Since the molecular mechanism of the upregulation of ATRX by ONC201 is not clear at present, we proceeded to study EZH2 expression. The rationale is that MYCN upregulates EZH2, leading to the inactivation of a tumor suppressor program in neuroblastoma [18], while ATRX in-frame fusion neuroblastoma is sensitive to EZH2 inhibition [19]. The results of our study reveal that ONC201 did not change the expression of EZH2 protein. Knock-down of EZH2 also failed to change the ATRX protein expression (Appendix A).

Finally, we also evaluated the potentially toxic effects of ONC201 on the mice by measuring the body weight, organ weight, and histology of two major organs, the liver and kidney, which showed no significant difference between groups (Appendix A).

Taken together, ONC201 achieves the significant tumor inhibitory effect through interrupting the mitochondria function and modulating the expression of ATRX and MYCN, the two critical clinical risk factors for patients with NB.

## 3. Discussion

The present study first confirms a role for ClpXP protease in the differentiation of human NB tissues. Using ONC201 to suppress ClpXP protease, we observed significant decreases in mitochondrial proteases ClpP and ClpX in NB cells, regardless of the *MYCN* amplification status, which is associated with impaired mitochondrial respiratory chain function. ONC201 alone can induce a state of energy depletion, increased ROS, and decreased mitochondrial membrane potential in NB cells. The overall cascade effect is significantly enhanced cytochrome c accumulation in the mitochondria, release into the cytosol, and consequent suppression of NB cell growth through mitochondria-mediated apoptosis, which is particularly significant in *MYCN*- amplified NB cells. The latter is also vulnerable to glycolytic inhibition by 2DG than *MYCN*-nonamplified NB cells [1,2]. We also found that MYCN is suppressed in *MYCN*-amplified NB cells, while ATRX is activated and accumulates in the nucleus after suppression of ClpP and ClpX with ONC201 or by siRNA targeting the genes in both NB cell types. The findings are critical since inactivation of ATRX has been correlated with a high risk of the disease and poor prognosis [20,21]. Our findings indicate that ATRX could be reactivated by ONC201 treatment or by suppressing of ClpP/ClpX expression to improve the survival of the patients with high-risk *MYCN*- amplified NB.

We found that the highest immunoreactive staining of both ClpP and ClpX proteins occurred in immature or mature ganglion cells rather than more malignant neuroblasts in human NB tumors. The expression is also low in the Schwannian-stroma-dominant cell type, as they are deficient in ganglion cells and also a favorable histological finding, according to the Shimada classification [22]. The implication of ClpP and ClpX in neuronal differentiation and maturation in NB seems contrary to previous studies reporting that ClpXP is overexpressed in hematologic malignancies and solid tumors, and is necessary for the viability of a subset of tumors [7]. Both inhibition and hyperactivation of ClpXP lead to impaired respiratory chain activity and causes cell death in cancer cells [7]. The data provided in our study clearly show that ONC201 suppresses ClpXP and disrupts mitochondrial function in vitro, but repeated treatment with a high dose of ONC201 in *MYCN*-amplified NB xenograft promote the expression of immunoreactive ClpP and ClpX proteins. The latter is consistent with several other cancers, such as acute myeloid leukemia, in which ONC201 functions as an activator of ClpXP [9,23].

Respiratory chain subunits have been identified as putative ClpP substrates, while the inhibition of ClpP leads to the accumulation of degraded or misfolded subunits. Our findings of notable downregulations of SDHB and NDUFS1 are consistent with a study reporting that ClpP hyperactivation induces lethality in leukemias and lymphomas, while selective proteolysis of subsets of the mitochondrial proteome involving mitochondrial respiration and oxidative phosphorylation are responsible [23]. As one of the subunits of mitochondrial respiratory chain complex II, SDHB has been shown to significantly increase the carcinogenesis of oral squamous cell carcinoma [24]. By contrast, SDHB has been reported to suppress tumorigenesis in clear cell renal cell carcinoma, while decreased SDHB accelerates growth in hepatocellular carcinoma (HCC) [25,26], which is ascribed to a switch from aerobic respiration to glycolysis in HCC. In this study, ONC201 decreased the expression of SDHB with the interruption of aerobic respiration in NB cells, eventually leading to NB cell death.

Repression of genes, including NDUFS1, by a sulfonamide anticancer agent, indisulam, has been shown to improve the survival of patients with metastatic melanoma [27]. Granzyme B, a caspase and cytotoxic lymphocyte protease, can increase ROS in target cells by directly cleaving complex I subunits, including NDUFV1, NDUFS1, and NDUFS2, to promote apoptosis [28]. Interestingly, oncostatin M, a pleiotropic cytokine belonging to the IL-6 family, may suppress NDUFS1/2 and improve the glioblastoma response to ionizing radiation, thereby prolonging lifespan [29]. The above findings are consistent with our results of a profound decrease in NDUFS1 using ONC201, leading to increased ROS and promotion of apoptosis in susceptible NB cells.

ROS increases were detected in the NB cells treated with ONC201. ONC201-induced reduction in ClpP has been shown to blunt UPR^mt^ induction, leading to increased generation of ROS, and decreased membrane potential [14]. Interestingly, both superoxide and hydrogen peroxide production, and mitochondrial ROS generation are highest in SK-N-AS, which is also known to have a high expression of c-Myc. The latter is known to regulate ROS generation along with HIF-1 alpha [30], which may be ascribed to the c-Myc induction of nuclear-encoded mitochondrial gene expression and mitochondrial biogenesis [31]. Gene expression analysis has revealed that MYCN is associated with increased ROS, downregulated mitophagy, and poor prognosis [32]. Forced overexpression of MYCN in neural crest progenitor cells enhances glutaminolysis, leading to ROS production and rendering NB cells sensitive to ROS augmentation [33].

Downregulation of ClpXP by ONC201 may be responsible for a decrease in ClpP in myoblasts, the mouse islet β-cell line (Min6), and human trophoblast cells, which are all associated with decreased MMP [14,34,35]. Contrary to a previous study reporting that the downregulation of ClpP in muscle cells impairs myoblast differentiation [14], ONC201 treatment or siRNA targeting ClpP and ClpX in our study significantly increased ATRX expression in the nucleus of BE(2)M17. ATRX has been implicated in neuronal differentiation, which may be impaired by the silencing of ATRX [36]. ATRX as a tumor suppressor has been associated with protection from DNA replication stress through a resolution of difficult-to-replicate G-quadruplex (G4) DNA structures [37]. Meanwhile, ATRX mutation has been regarded as an unfavorable histology [38]. Our finding of an increased expression of ATRX in the nucleus is of interest and significant. A recent report of a case with ATRX mutation associated with complex I deficiency suggests that target genes of the ATRX protein include those responsible for mitochondrial function [39]. ONC201 and ClpXP protease target mitochondria, which concurrently upregulate ATRX expression in the nucleus, indicating that some form of feedback loop exists in this particular scenario. A previous study revealed that MYCN upregulates EZH2, leading to the inactivation of a tumor suppressor program in neuroblastoma [18]; meanwhile ATRX in-frame fusion neuroblastoma is sensitive to EZH2 inhibition [19]. Our study, using ONC201 or siRNA targeting EZH2, reveals that even using higher doses that could significantly decrease EZH2 protein expression, there is little change in ATRX expression. Our results fail to duplicate that found in glioma or other cancers [40,41]. Obviously, the genetic background in our NB cell models could be responsible for their difference from diffuse midline glioma. Special manipulation, such as ATRX in-frame fusion [19], may be required for this purpose. We will continue to solve this problem in our future study.

*MYCN* gene amplification is one of the most important prognostic markers for high-risk NB patients. The reported prevalence of NB patients with *MYCN* gene amplification is around 20–30%, with poor overall survival rate of less than 50% [42]. A recent report on targeting of the *MYCN* gene by genetic deletion using DNA alkylating agent, MYCN-A3, induced apoptosis in *MYCN*-amplified cells, but not in nonamplified cells [43]. In our present study, we found that ONC201 treatment could significantly inhibit the protein expression of MYCN in *MYCN*-amplified NB cells with a promising tumor suppressor effect in xenograft tumors. Although the exact mechanism is needed to be elucidated, the tumor inhibitory effect is encouraging for future application to treat high-risk NB patients.

One limitation of the current study is that we did not investigate how ONC201 decreases the expression of MYCN in MYCN-amplified NB cells. Similar findings are shown in a study using MYCN-amplified IMR-32 and MYCN-nonamplified SK-N-SH NB cells to address the issues of differential tumorigenic protein expression [44]. They also failed to verify the mechanism explaining how ONC201 and ONC206 accomplish the specific aim of decreasing the expression of MYCN and several other tumorigenic proteins. Future studies are necessary to clarify the issue. 

## 4. Materials and Methods

### 4.1. Cell Cultures and Reagents

Human NB cell lines SK-N-AS and BE (2)-M17 were purchased from the American Type Culture Collection (Manassas, VA, USA). All cell lines were maintained in Dulbecco’s modified Eagle’s medium (DMEM) (Thermo Fisher Scientific, Waltham, MA, USA) and supplemented with 10% heat-inactivated fetal bovine serum (FBS; Thermo Fisher Scientific, Waltham, MA, USA), GlutaMAX (Thermo Fisher Scientific, Waltham, MA, USA), nonessential amino acids (Thermo Fisher Scientific, Waltham, MA, USA), and an antibiotic–antimycotic (Thermo Fisher Scientific, Waltham, MA, USA) in a 5% CO_2_ humidified incubator at 37 °C. ONC201 (SML1068), DHE (309800), and Resazurin sodium salt (199303) were purchased from Sigma-Aldrich (St. Louis, MO, USA). Cleaved caspase 3 (Asp175) and N-Myc (9405S) antibodies were purchased from Cell Signaling Technology (Danvers, MA, USA). ClpX (ab168338), ATRX (ab97508), and EZH2 (ab186006) antibodies were purchased from Abcam (Cambridge, MA, USA). β-actin (sc-8432), ClpP (sc-271284), SDHB (sc-271548), VDAC1 (sc-390996), Lamin b (sc-6216), and NDUFS1 (sc-271510) were purchased from Santa Cruz Biotechnology (Santa Cruz, CA, USA). A-tubulin (GTX112141) was purchased from GeneTex (Hsinchu, Taiwan).

### 4.2. Neuroblastoma Xenograft Animal Model

Four-week-old male nonobese diabetic/SCID (NOD/SCID, NOD.CB17-Prkdcscid/NcrCrl) mice were purchased from the National Laboratory Animal Center (Taipei, Taiwan). All animal procedures in this study are approved by the Institute of Animal Care and Use Committee of Kaohsiung Chang Gung Memorial Hospital, Taiwan (IACUC No. 2019091805). To induce human neuroblastoma, SK-N-DZ cells were subcutaneously inoculated into the right flank of mice (1 × 10^7^ cells in 0.1 mL of medium). After implantation for 8 days, tumor-bearing mice were randomly divided into control (*n* = 3), ONC201 (10 μg/g), and ONC201 (50 μg/g) groups (*n* = 5), respectively. The mice then received intraperitoneal injections of the indicated treatment once weekly for 4 weeks. At the end of the experiment, mice were sacrificed, and the tumors were harvested and weighed using an electronic microbalance.

### 4.3. Cell Survival Assessment

The Resazurin reduction test was utilized to detect the cell survival. Briefly, NB cells were seeded in a 96-well culture plate (1.5 × 10^4^ cells/well) overnight. Then, cells were treated with indicated concentrations of ONC201 for 48 h and 96 h. At the end of the experiment, cells were supplemented with 20 μL per well of Resazurin dye (Sigma-Aldrich, St. Louis, MO, USA) and incubated for another 3 h at 37 °C. The optical density of resorufin was measured at 570 nm using a 96-well spectrophotometric plate reader (Hidex Sense, Turku, Finland). 

### 4.4. Flow Cytometry to Detect Cell Death

Cell death was detected by trypan blue solution (Thermo Fisher Scientific, Waltham, MA, USA). After ONC201 (5 μM) treatment, cells were dissociated from the plate with trypsin-EDTA and then stained with trypan blue solution for 5 min. The percentage of cell death was detected by FACS caliber 101 flow cytometer (BD Biosciences, San Jose, CA, USA) and analyzed using winMDI software. 

### 4.5. Terminal Deoxynucleotidyl Transferase dUTP Nick End Labeling (TUNEL) Assay 

ONC201-induced NB cell death was detected by enzyme labeling of DNA strand breaks using a TUNEL assay (In Situ Cell Death Detection Kit; Roche, Germany) according to manufacturer’s instructions. Briefly, cells were fixed with 4% paraformaldehyde for 10 min and then incubated with a mixture of terminal deoxynucleotidyl transferase and fluorescein-dUTP at 37 °C to label free 3′OH ends of DNA. After TUNEL staining, cells were washed with PBS and counterstained with DAPI Fluoromount-G (SouthernBiotech, Birmingham, AL, USA) for 10 min and viewed under a fluorescent microscope. For quantifying the immunostaining of TUNEL, each group was randomly captured by microscopy for three independent fields. The apoptotic percentage was estimated by normalizing the number of TUNEL-positive cells to the total number of DAPI-positive cells and calculated from three independent fields.

### 4.6. Isolation of Mitochondrial and Cytosolic Fractions

After ONC201 treatment for 24 h, cells were harvested and resuspended in 1.2 mL RSB Hypo Buffer (10 mM Tris-HCl, pH 7.5, containing 10 mM NaCl and 1.5 mM MgCl2) for 10 min on ice. Subsequently, cells were homogenized using a 1 mL syringe to pass the cell suspension through a 27-gauge needle 10 times, and then added to 0.8 mL 2.5X MS homogenization buffer (12.5 mM Tris-HCl, pH 7.5, containing 2.5 mM EDTA, 525 mM mannitol, and 175 mM sucrose). Nuclei and intact cells were removed by centrifugation at 3000× *g* rpm for 5 min at 4 °C. The supernatants (cytoplasmic fraction) were recentrifuged at 13,000 rpm for 20 min at 4 °C to move pellets, while the pellets containing mitochondria were washed with 1X MS homogenization buffer (5 mM Tris-HCl, pH 7.5, containing 1 mM EDTA, 210 mM mannitol and 70 mM sucrose) twice at 13,000 rpm for 10 min at 4 °C, and then resuspended in PRO-PREP^TM^ Protein Extraction Solution (Intron Biotechnology Inc., Seongnam, Republic of Korea) for 20 min on ice. The separation efficiency was determined by Western blot.

### 4.7. Isolation of Nuclear and Cytosolic Fractions

Nuclear and cytosolic fractions were extracted according to L&W nucleus/cytoplasm fractionation protocol [45]. Cells were harvested and resuspended in 0.5 mL hypotonic solution (20 mM Tris-HCl pH 7.4, 10 mM KCl, 2 mM MgCl_2_, 1 mM EGTA, 0.5 mM DTT, 0.5 mM PMSF) containing 0.1% NP-40 and incubated for 3 min on ice. Subsequently, cells were homogenized using a 1 mL syringe to pass the cell suspension through a 27-gauge needle 10 times, and then centrifugation at 3000× *g* rpm for 5 min at 4 °C. The supernatants (cytoplasmic fraction) were recentrifuged at 13,000 rpm for 10 min to move pellet debris, while the pellets containing nuclei were washed with isotonic buffer (20 mM Tris-HCl pH 7.4, 150 mM KCl, 2 mM MgCl_2_, 1 mM EGTA, 0.5 mM DTT, 0.5 mM PMSF) three times, and then resuspended in PRO-PREP^TM^ Protein Extraction Solution (Intron Biotechnology Inc., Seongnam, Republic of Korea) for 20 min on ice. The supernatants (nuclear fractions) were obtained by centrifugation at 13,000× *g* rpm for 10 min at 4 °C. The separation efficiency was determined by Western blot.

### 4.8. Mitochondrial Bioenergetics Analysis 

The rate of oxygen consumption (OCR) and extracellular acidification rate (ECAR) in NB cells were analyzed using a Seahorse XF24 extracellular flux analyzer (Seahorse Bioscience Inc.; Chicopee, MA, USA) [46]. Initially, cells were seeded in Seahorse cell culture 24-well plates (2 × 10^4^ cells/well) overnight, and then treated with ONC201 (5 μM) for 48 h. After washing with sodium-bicarbonate-free DMEM medium, cells were refreshed with 500 μL of medium for further examination. The basic OCR was measured four times and plotted as a function of the cells under basal conditions; the inhibitors, including oligomycin (1 μM), FCCP (0.5 μM), and rotenone (1 μM), were added sequentially before experimental analysis. The basal ECAR was measured after injections of glucose (10 mM) and oligomycin (1 μM). At the end of recording, cells were collected and counted using a trypan blue exclusion assay. The OCR and ECAR values were normalized to total protein levels in individual wells using the BCA protein assay (Thermo Fisher Scientific, Waltham, MA, USA).

### 4.9. Measurement of ROS Generation

The intracellular ROS generation was measured by flow cytometry using DHE staining. Cells were treated with ONC201 (5 μM) for 48 h, and these cells were incubated with DHE (10 μM) for 30 min at 37 °C. The ROS production was further measured using a FACS caliber 101 flow cytometer (BD Biosciences, San Jose, CA, USA) and analyzed using winMDI software. For the detection of mitochondrial superoxide anions, the MitoSox Red reagent (Invitrogen; Carlsbad, CA, USA) was applied according to the manufacturer’s protocol. Briefly, at the indicated time points after treatment, cells were incubated with MitoSox Red reagent (5 μM) for 30 min at 37 °C. The cells were then collected, washed twice with PBS, and finally resuspended in a flow tube with 1 mL PBS. The fluorescent signal of cell suspension was then measured using a FACS caliber 101 flow cytometer (BD Biosciences, San Jose, CA, USA) and analyzed using winMDI software. 

### 4.10. Measurement of Mitochondrial Membrane Potential 

Mitochondrial membrane potential (MMP) was determined by Tetramethyl rhodamine methyl ester (TMRM; Sigma-Aldrich, USA) staining. Briefly, at the indicated time points after treatment, cells were incubated with TMRM (100 nM) for 20 min at 37 °C in the dark, washed twice with ice-cold PBS, and finally resuspended in l ml PBS. Mitochondrial permeability transition was measured immediately by FACS caliber 101 flow cytometer (BD Biosciences, San Jose, CA, USA) and analyzed using winMDI software.

### 4.11. Measurement of ATP Content

The cellular ATP concentrations were measured using the ATP Colorimetric/Fluorometric Assay Kit (BioVision, Inc., Milpitas, CA, USA) according to the manufacturer’s instructions. Briefly, after treatment with ONC201 (5 μM) for 48 h, cells (4 × 10^5^) were collected and resuspended in a reaction buffer containing 20 mM glycine, 50 mM MgSO_4_, and 4 mM EDTA. Samples were boiled for 2 min at 100 °C and centrifuged for 5 min at 3000× *g* rpm. Supernatants were collected, and 50 μL of each sample was mixed with 50 μL of an ATP solution for fluorescence readings at Ex/Em535/587 nm using a fluorescence meter (FLUOstar OPTIMA; BMG Labtech, Ortenberg, Germany). The level of ATP production was determined from a standard curve constructed with 10–100 pmol ATP.

### 4.12. Immunofluorescence Analysis

NB cells were plated overnight on 6-well culture dishes containing sterilized coverslips. After treatment for 48 h, cells were fixed with 4% (*w*/*v*) paraformaldehyde and permeabilized with PBS containing 0.1% (*w*/*v*) Triton X-100 and 2% (*w*/*v*) BSA at room temperature for 10 min. Cells were labeled with indicated primary antibodies, followed by Alexa Fluor-conjugated secondary antibody (Invitrogen; Carlsbad, CA, USA). Cells were then washed with PBS and mounted in DAPI Fluoromount-G (SouthernBiotech, Birmingham, AL, USA). Labeled cells were visualized with LSM510 (Carl Zeiss, Thornwood, NY, USA).

### 4.13. Western Blot Analysis

NB cells were treated with ONC201 (5 μM) for 48 h before protein analysis. Cell lysates were separated by SDS-PAGE and transferred to a polyvinylidene difluoride (PVDF) membrane. The PVD membrane was blocked with 5% milk in TBS-T for 1 h, then incubated with specific primary antibodies and secondary antibodies conjugated with HRP (1:10,000 dilutions in 5% milk) for 1 h, respectively. The membrane signals were analyzed using an AutoChemi image system (UVP), or exposed to Fuji medical X-ray film, followed by quantification with Alpha View SA 3.4.0 (ProteinSimple, San Jose, CA, USA).

### 4.14. Immunohistochemistry Analysis and Scoring

For the analysis of ClpXP expression in human NB tissues and ATRX profiles in neuroblastoma tissues, the paraffin sections were deparaffinized, blocked with 3% hydrogen peroxide for 10 min, and subjected to antigen retrieval by microwave heating in 0.01 M citrate buffer for 15 min. The slides were then washed twice with PBS, incubated with primary antibodies, followed by incubation with the polymer conjugated with peroxidase for 30 min using a polymer detection system (Zymed Laboratories, San Francisco, CA). Finally, the color was developed with 3, 3-diaminobenzidine (DAB; Sigma, St. Louis, MO). The slides were counterstained with Gill’s hematoxylin, dehydrated, and mounted before microscopic reading. The intensity of positively stained tumor cells was scored as 0 = none; 1 = weak; 2 = intermediate; and 3 = strong. For quantification of ClpXP and ATRX staining intensity, we randomly selected five high-power (×400) fields for each section to evaluate each sample. The intensity of positively stained tumor cells was scored as 0  =  none; 1  =  weak; 2  =  intermediate; and 3  =  strong. The proportion of each intensity score was further scored as 0  =  no positive cells; 1  =  0–20%; 2  =  21–50%; 3  =  51–80%; 4  =  81–100%. Both scores were multiplied and summed to produce a final immunoreactivity score, ranging from 0 to 12.

### 4.15. RNA Interference

For knocking down the gene expression of ClpXP protease, cells were transfected with control siRNA (D-001810-10-50; Dharmacon, Lafayette, Co, USA), siClpP (L-005811-00-0020; Dharmacon, Lafayette, Co, USA) and siClpX (L-008763-00-0020; Dharmacon, Lafayette, Co, USA) for 4 h, respectively, using Lipofectamine 3000 Reagent (Thermo Fisher Scientific, Waltham, MA, USA) according to the manufacturer’s instructions, and then incubated with complete medium for 48 h before the ensuing experimental analyses.

### 4.16. Statistical Analysis

Data are presented as the mean ± standard deviation of three independent experiments, unless otherwise indicated. The statistical analysis was performed using GraphPad Prism 8.0 software (GraphPad Software, San Diego, CA, USA). One-way ANOVA and post hoc multiple comparison via the Tukey test were used to compare multiple groups, and *t*-tests were used for two-group comparisons. A *p*-value less than 0.05 was considered statistically significant.

## 5. Conclusions

The dysregulation of mitochondria, suppression of MYCN, reactivation of ATRX, and the implication of ClpP and ClpX in differentiation are key processes involved in the particular process of ONC201 treatment in NB cells. ONC201, as a single-agent therapy, can suppress *MYCN*-amplified NB xenograft growth. Future therapeutic application in NB, in combination with other agents, is promising. The major finding of our study is summarized and illustrated in Figure 7.

## Figures and Tables

**Figure 1 ijms-24-01649-f001:**
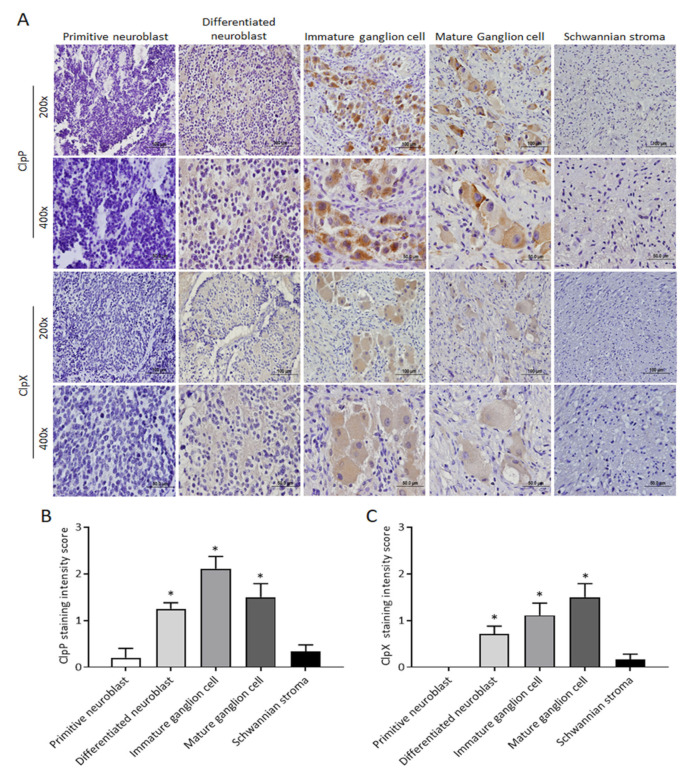
Comparison of ClpP and ClpX expressions in subtypes of human NBs: (**A**) Representative profiles of ClpP and ClpX immunostaining in differentiated NBs. (**B**,**C**) Histological scoring for ClpP and ClpX intensity in NBs tissues. Data in each bar chart are represented as mean ± SD. Primitive neuroblast (*n* = 4), differentiated neuroblast (*n* = 12), immature ganglion cell (*n* = 10), mature ganglion cell (*n* = 4), and Schwannian stroma (*n* = 12). Upper panel, ×200 magnification, scale bar 100 μm; lower panel, ×400 magnification, scale bar 50 μm. * *p* < 0.05.

**Figure 2 ijms-24-01649-f002:**
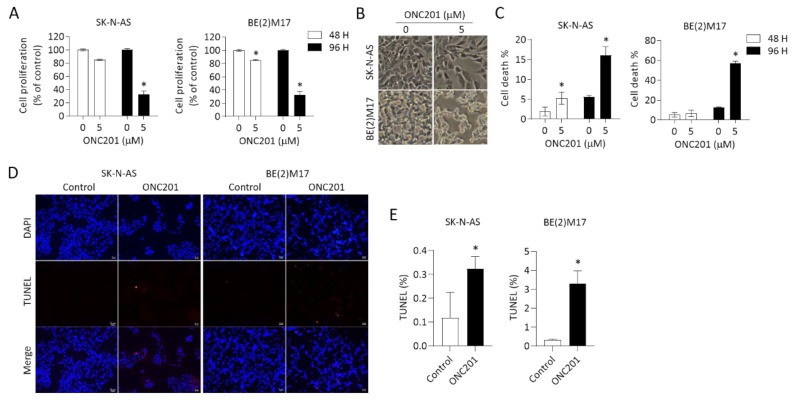
Effects of ONC201 on cell survival in NB cells: (**A**) SK-N-AS and BE(2)M17 cells were treated with ONC201 (5 μM) for 48 h and 96 h. The cell survival was detected by Resazurin assay. (**B**) Representative phase contrast images demonstrating cell density and morphological changes in NBs exposed to ONC201 for 48 h. Scale bar 50 μm. (**C**) After incubation with ONC201 (5 μM) for 48 h, cells were collected and stained by trypan blue solution. The percentage of trypan-blue-positive cells was quantified using flow cytometry. (**D**,**E**) Representative profiles of TUNEL staining (red) in control and ONC201-treated NB cells. DAPI (blue) was used as a nuclear counterstain. Scale bar 20 μm. (**D**) The apoptotic cells are expressed as the percentage of the number of TUNEL-positive cells over the number of DAPI-positive cells. All data are expressed as mean ± SD from triplicate experiments. * *p* < 0.05.

**Figure 3 ijms-24-01649-f003:**
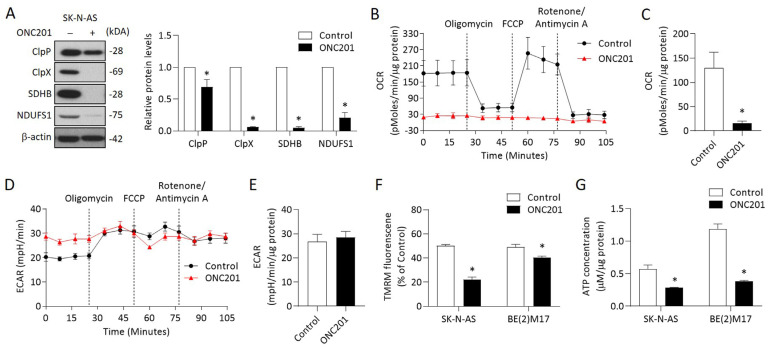
Effects of ONC201 on mitochondrial functions in NB cells: (**A**) The relative protein levels of ClpP, ClpX, SDHB, and NDUFS1 were determined by Western blot and quantified by Image Pro-plus analysis software. (**B**,**C**) OCR in ONC201-treated cells were detected by the Seahorse XF24 analyzer in the absence or presence of oligomycin (1 μM), FCCP (0.5 μM), and rotenone (1 μM) by counting 1 × 10^5^ cells. (**D**,**E**) Profiling of ECAR in control and ONC201-treated cells was measured by the Seahorse XF24 analyzer. (**F**) Cells were labeled with TMRM (100 nM) to examine mitochondrial membrane potential and analyzed by flow cytometry. (**G**) Total cellular ATP levels were measured using the luciferase-based luminescence assay kit. All data were obtained from three independent experiments, and are expressed as mean ± SD. * *p* < 0.05.

**Figure 4 ijms-24-01649-f004:**
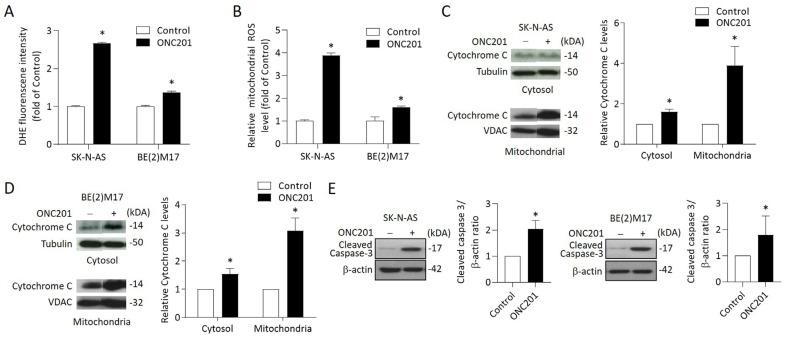
Effects of ONC201 on mitochondria-mediated apoptosis signaling in NB cells: (**A**) The intracellular ROS production of ONC201-treated cells was detected by DHE staining. The relative intensity of DHE fluorescence was analyzed by flow cytometry. (**B**) Mitochondrial ROS was detected by MitoSOX Red and immediately analyzed by flow cytometry. Data are expressed as fold change compared with control group. (**C**,**D**) To detect cytochrome C release, cytosolic and mitochondrial fractions of NB cells were subjected into 15% SDS-PAGE and probed with anti-cytochrome c antibody by Western blot analysis. Alpha-tubulin and VDAC1 were used as the cytosolic and mitochondrial internal control, respectively. (**E**) The protein levels of cleaved caspase 3 were detected by Western blot. The relative protein ratio was normalized with β-actin using Image Pro-plus analysis software. All data were obtained from three independent experiments, and are expressed as mean ± SD. * *p* < 0.05.

**Figure 5 ijms-24-01649-f005:**
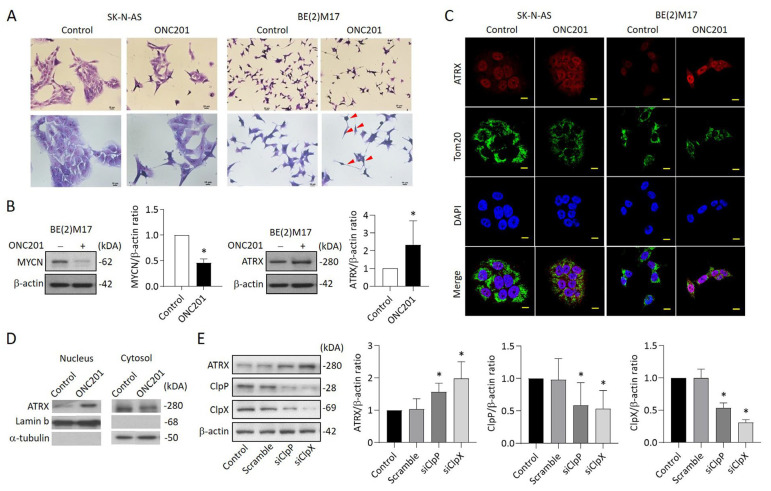
Effects of ONC201 on neurite outgrowth and the express profiles of ATRX in NB cells: (**A**) Representative of Giemsa staining in ONC201-treated SK-N-AS and BE(2)M17 cells. Scale bar 20 μm (**upper panel**) and 10 μm (**lower panel**), respectively. Red arrow indicates the neurite extension. (**B**) Western blot analysis of MYCN and ATRX protein in ONC201-treated BE(2)M17 cells. The relative protein ratio was normalized with β-actin using Image Pro-plus analysis software. (**C**) Representative images demonstrating ATRX enhancement (red) in the nucleus of SK-N-AS and BE(2)M17 cells when exposed to ONC201 (5 μM) for 48 h. The mitochondria were labeled with anti-TOM20 antibody (green). The nuclei were labeled with DAPI (blue). Scale bar 10 µm. (**D**) Western blot analysis of ATRX protein in nuclear and cytosolic fractions in BE(2)M17 cells. The relative ATRX ratio was normalized with lamin b and alpha-tubulin, respectively, and expressed as mean ± SD. (**E**) Analysis of ATRX expression in siClpP- and siClpX-treated SK-N-AS cells by Western blot. The relative protein ratio was normalized with β-actin, and is expressed as mean ± SD. All data were obtained from four to six independent experiments. * *p* < 0.05.

**Figure 6 ijms-24-01649-f006:**
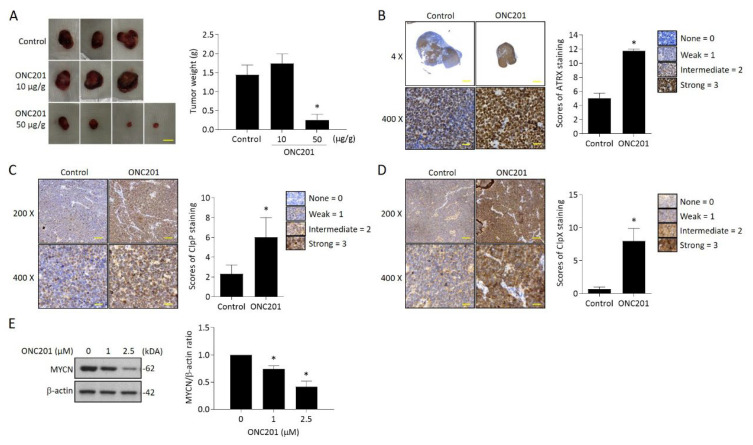
Effect of ONC201 administration on tumor growth in NB xenograft animal model and ClpXP and ATRX expression in tumor tissues: (**A**) Images of the resected tumors in the three groups. Tumor weight (g) was measured at sacrifice by microbalance and expressed as mean ± SD. Control = 3, treatment = 5. Scale bar 1 cm. (**B**) Representative pictures of ATRX expression in control and ONC201-treated neuroblastoma tissues. Scale bar 100 μm (**upper panel**) and 30 μm (**lower panel**), respectively. The percentage of nuclear ATRX-positive cells was calculated from five random images at 400x magnification per tissue, and are expressed as mean ± SD (*n* = 3). (**C**,**D**) Immunohistochemistry analysis of ClpP and ClpX expression in controls and ONC201-treated neuroblastoma tissues. Scale bar 30 μm. The intensities of ClpP and ClpX staining were calculated from five random images at 400× magnification per tissue, and are expressed as mean ± SD (*n* = 3). (**E**) Analysis of MYCN expression in ONC201-treated BE(2)M17 cells by Western blot. The relative protein ratio was normalized with β-actin using Image Pro-plus analysis software. * *p* < 0.05.

**Figure 7 ijms-24-01649-f007:**
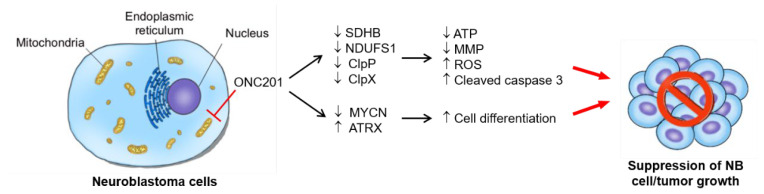
The proposed working model for interruption of mitochondrial functions, induced apoptosis, and differentiation in NB cells: ONC201 treatment targets mitochondria to perturb mitochondrial functions, e.g., the treatment disrupts the mitochondrial ClpXP complex, impairs respiratory chain activity, and leads to mitochondrial function loss and ROS accumulation. The enhancement of ROS production finally induces apoptosis in NB cells. In addition to cell death, ONC201 treatment also promotes the nuclear expression of ATRX, which supports neuronal differentiation and the suppression of NB growth.

## Data Availability

The data presented in this study are available on request from the corresponding author. The data are not publicly available due to privacy.

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
