# Peer review of "ONC201 Suppresses Neuroblastoma Growth by Interrupting Mitochondrial Function and Reactivating Nuclear ATRX Expression While Decreasing MYCN"

_ijms, 2023, doi:10.3390/ijms24021649_

Round 1
Reviewer 1 Report (Previous Reviewer 4)
Glad to be the reviewer again for this paper, the authors addressed part of the comments that I questioned last time.
I still have some questions this time.
In fig4E, total ATP were evaluated for the mitochondrial function which is not specific. Besides the mitochondrial, other processes also produce the ATP. Mitochondrial isolation could be better for this experiment.
States the statistical method for the comparison of different groups in all statistical pictures, like the T test/ one-way anova etc.
You did the combination study in vitro with the 2DG and ONC201, why not evaluate it in vivo?
Also we did not see the ClpP and ClpX in vivo study by IHC or WB.
Please check the grammar or Misspell in this manuscript for example the misspell of “control” in Fig. S4.
Author Response
We really appreciate all the constructive suggestions and comments from the respective reviewers. However, we could not complete all the experiments to answer all the questions in a limited period. We will re-organize our research to get more scientific experiment data according to the suggestions and comments.
Thanks again for your time and the excellent comment.

Reviewer 2 Report (New Reviewer)
The aim of this study by Chung et a. is to investigate the underlying mechanism of ONC201 against neuroblastoma. The authors found that ONC201 induced apoptosis through the downregulation of Clpx and Clpp and upregulation of ATRX. The manuscript is well written. However, there are some concerns need to be addressed.
Major concerns
1. ONC201 targeting on CLPP have been well discovered in glioma. The novelty of this study is to identify the fact that ONC201 upregulates ATRX and downregulates N-MYC in the N-MYC-expressed neuroblastoma line. The xenograft data showed that ONC201 significantly inhibits tumor growth. However, cell culture data ONC201 alone has less effect on cell proliferation. The trypan blue staining showed a significant increase which was only 5%. The great increase was only observed in the combination of ONC201 and 2DG. The tumor suppression may be caused by other mechanisms rather than CLPP inhibition.
2. The results from Western blotting analysis in Figure 5B showed that ONC201 alone was unable to increase ATRX in both neuroblastoma lines. However, the results of immunofluorescence assay in Figure 5C indicated that ONC201 increases ATRX. What could be the causes leading to the difference between two assay methods.
3. What could be the cause to increase nuclear ATRX by ONC201?
4. The treatment of ONC201 alone can downregulate the Clpx and Clpp expression (Figure 3A). However, ONC201 alone was unable to increase the total ATRX expression (Figure 5B).
5. ONC201 can downregulate CLPP and increase ATRX. The knockdown of Clpx and Clpp leads to an increase of ATRX. It is difficult to decide that ONC201 upregulates ATRX through the downregulation of Clpx and Clpp. Whether the upregulation or overexpression of CLPP downregulate ATRX?
Minor concerns
1. Please show the calculation method how to get the values in Figure 3C and 3D. Did Figure 3C present the comparison in basal, maximal respiratory, or others? Did Figure 4 present the glycolysis, glycolytic capacity, or others? Please specify the meaning of two figures. Please also show the ECAR picture by seahorse assay.
2. Please unify the TMRE or TMRM, including the labels of Y-axis in figure 3F.
3. Please show the isolation method of mitochondria.
4. In Figure 4C, which VDAC was showed in blot, VDAC1/2/3 or total VDAC. Which tubulin was used for the loading control of cytosol, alpha- or beta-tubulin?
5. In Figure 4D, the blot showed that both 2DG and ONC201 significantly increase cytochrome C in cytosol fraction despite that there was no significance in band quantification.
6. In Figure 5C, please show the DAPI stain.
7. In Figure 5D, the nuclear marker is lamin b, not lamine b.
8. The quantification of ATRX band in nucleus fraction was significantly increased by 2DG/ONC201. However, it is difficult to identify the increase based on the presented blot.
Author Response
We really appreciate all the constructive suggestions and comments from the respective reviewers. However, we could not complete all the experiments to answer all the questions in a limited period. We will re-organize our research to get more scientific experiment data according to the suggestions and comments.
Thanks again for your time and the excellent comment.

This manuscript is a resubmission of an earlier submission. The following is a list of the peer review reports and author responses from that submission.
Round 1
Reviewer 1 Report
The present manuscript is focused on the characterization of the activity of ONC201 as therapeutic agent in neuroblastoma. The authors provided a wide range of assay aimed at showing the outcomes of ONC201 treatment both in vitro and in vivo.
Despite the effort of providing a broad characterization of ONC201 mechanism of action, some among the explored aspects are still not clear and requires further experimental investigations.
Comments:
1. Is ONC201 from the imipridone drug class? Is the small molecule originally identified as a TNF-related apoptosis inducing ligand (TRAIL)-inducing compound and phase II drug for glioblastoma?
2. Could you provide direct evidence for the ONC201 activity as mitochondrial ClpXP activator/suppressor? Could you explain its MoA? Are there any other ONC201 known targets?
3. Fig. 2A-C. It seems that ONC201 compound alone poorly affect cell proliferation, whereas 2DG strongly block it (fig. 2A). There is a little additive effect when the two drugs are combined. On the other hand ONC201 alone affect viability (fig.2C) more then 2DG and the combo effect is sensitive enhanced. Could you elaborate on that?
4. Figure 2 caption. It seems that the image and the caption are in disagree in respect to ONC201 compound concentration, could you double check?
5. Could you show protein level for ClpP and ClpX via immunoblot for SK-N-AS cellsv and in BE(2)M17 cells? Could you include in the panel other NB cells and relevant control cell with both positive and negative reference?
6. What happens when you knock down ClpP or ClpX and look at the same readout generated with ONC201 (as in figure 2, 3 or 4)?
7. The Company “Chimerix” define ONC201 as “the founding member of the imipridone class of anti-cancer small molecules which selectively targets Dopamine Receptor D2 (DRD2) and ClpP” (ref. https://www.chimerix.com/our-pipeline/imipridones/onc201/). Did you check for DRD2 expression in your neuroblastoma models?
8. 2DG. Its rapid metabolism and short half-life (according to Hansen et al., after infusion of 50 mg/kg 2-DG, its plasma half-life was only 48 min [117]) make 2-DG a rather poor drug candidate. Do you plan to test other compound with similar action or better PK properties?
Reviewer 2 Report
The paper has merit and is scientifically sound. However, a bit more clarity is needed in terms of what the authors are interested in – are they more interested in looking at the ClpPX multimeric complex as a whole or are they more interested in the individual component (ClpP and ClpX)? The introduction needs to provide more details and better rationale for why they are sometimes looking at the complex as whole vs as individual subunits.
Line 125 label for method is wrong “this section is not neuroblastoma xenograft animal model”
Line121 says they only used n=3 mice per group (need more mice)
Need a western blot for cell lines showing expression of ClpP or ClpX is present
Are the doses of ONC201 and 2DG being used clinically-relevant?
Line 289 has “archived” but should be “achieved”
For in vivo studies were there any toxicity issues. Please show body weights and/or histology of organs.
For in vivo studies it would be good to show some pharmacodynamic analysis (i.e. the drugs are targeting the protein of interest)
Reviewer 3 Report
The current manuscript entitled, ONC 201 suppresses neuroblastoma growth by interrupting mitochondrial function, reactivation of nuclear ATRX expression while decreasing MYCN is a well-designed, well-organised research paper. The authors did a great study here and provided enough experimental evidence to prove their hypothesis.
The abstract is well written.
The introduction part is good, but could be better by providing some more information on the background, hypothesis / research question with some more citations (if possible).
The materials and method part is mentioned in detail, however, please properly use superscripts and subscripts (for example, CO2, 107, 104 etc.)
Figure 1A, please enlarge the images to provide clearer images with more information. Could include a magnified view lane (like Fig-5A). Please show the changes in the images.
Figure 2A (Right panel), please show the significance (as shown in left panel) if this data is significantly different.
Figure 2B, Please show the changes in the images.
Figure 3, 4 are unnecessarily too small. Please enlarge the whole figure.
Figure 5A, please show the difference in the figure.
Figure 5C, please enlarge.
Please write a conclusion separately.
Reviewer 4 Report
In this manuscript the authors state that ONC201 could inhibit the Neuroblastoma progression both in vitro and in vivo. Furthermore, they employed the combination of 2DG strategy, which represents another mitochondrial metabolism pathway, and this showed more promising therapy effect for the Neuroblastoma. Some questions raised here.
1. A lot of combined study was performed but only ONC201 mentioned in the title and no combination related statement.
2. Be careful to check the manuscript before submitting, for example, the content is the same in line113 and line 125.The subtitles are not highlighted, line 186,195 etc.
3. 3 mice per group is not sufficient for the mice study. There is no combination study in vivo. Why only non- MYCN-amplified cells used for animal study?
4. N-MYC protein expression was inhibited in MYCN-amplified cells, what about the non- MYCN-amplified cells?
5. Is the ONC201 a mitochondrial ClpXP activator or suppressor(line 32), any reference?
6. In fig 5E after being treated with the specific siRNA CLpP, why did the CLpX also decrease? The same with CLpX siRNA.
7. We see the significance of DG and ONC201 combination effects in the SK-N-AS but no significant effect in BE(2)M17,if not, label the pvalue or show ns.
8. It looks like the ONC201 has better efficacy in the MYCN-non-amplified SK-N-AS cells than in MYCN-amplified BE(2)M17 cells. Why do you emphasize the MYCN in the title?
9. Some small errors need to be carefully checked.
Round 2
Reviewer 1 Report
Upon a round of revision the author did not improved sufficiently the level of the manuscript to fulfill the journal quality criteria.
I believe the manuscript is not enough solid to be suitable for publishing in this journal.
Author Response
Thanks for your excellent comments and suggestions.
Reviewer 2 Report
The manuscript entitled “ONC 201 suppresses neuroblastoma growth by interrupting mitochondrial function, reactivation of nuclear ATRX expression while decreasing MYCN,” has promise and merit for the scientific community. Though, I would like to point out that following revisions by the authors -no figures were present in the main manuscript. Figure legends were present in the back but no Figs were in the main part of the manuscript. Thus, making it very challenging for me to accept the paper. Furthermore, the authors did not respond to some of my comments from the first time the manuscript was reviewed. Please see below on the comments that were not addressed by the authors:
· Need a western blot for cell lines showing expression of ClpP or ClpX is present
· Are the doses of ONC201 and 2DG being used clinically-relevant?
· For in vivo studies were there any toxicity issues. Please show body weights and/or histology of organs.
· For in vivo studies it would be good to show some pharmacodynamic analysis (i.e. the drugs are targeting the protein of interest)
Reviewer 4 Report
The authors addressed all my concerns.
I have no other questions.
Author Response

(The authors gave the same response as above.)
